# Scoping and targeted reviews to support development of SPIRIT and CONSORT extensions for randomised controlled trials with surrogate primary endpoints: protocol

Anthony Muchai Manyara ![ORCID],[1] Philippa Davies,[2] Derek Stewart,[3] Valerie Wells,[1] Christopher Weir ![ORCID],[4] Amber Young ![ORCID],[2] Rod Taylor,[1,5] Oriana Ciani[6]

We wish to acknowledge our dear colleague Professor Amber Young who lost her brave battle with cancer and passed away on 17th September 2022.

For numbered affiliations see end of article.

**Correspondence to**
Anthony Muchai Manyara;
anthony.manyara@glasgow.ac.uk

## ABSTRACT

**Introduction** Using a surrogate endpoint as a substitute for a primary patient-relevant outcome enables randomised controlled trials (RCTs) to be conducted more efficiently, that is, with shorter time, smaller sample size and lower cost. However, there is currently no consensus-driven guideline for the reporting of RCTs using a surrogate endpoint as a primary outcome; therefore, we seek to develop SPIRIT (Standard Protocol Items: Recommendations for Interventional Trials) and CONSORT (Consolidated Standards of Reporting Trials) extensions to improve the design and reporting of these trials. As an initial step, scoping and targeted reviews will identify potential items for inclusion in the extensions and participants to contribute to a Delphi consensus process.

**Methods and analysis** The scoping review will search and include literature reporting on the current understanding, limitations and guidance on using surrogate endpoints in trials. Relevant literature will be identified through: (1) bibliographic databases; (2) grey literature; (3) handsearching of reference lists and (4) solicitation from experts. Data from eligible records will be thematically analysed into potential items for inclusion in extensions. The targeted review will search for RCT reports and protocols published from 2017 to 2021 in six high impact general medical journals. Trial corresponding author contacts will be listed as potential participants for the Delphi exercise.

**Ethics and dissemination** Ethical approval is not required. The reviews will support the development of SPIRIT and CONSORT extensions for reporting surrogate primary endpoints (surrogate endpoint as the primary outcome). The findings will be published in open-access publications.

This review has been prospectively registered in the OSF Registration DOI: 10.17605/OSF.IO/WP3QH.

## INTRODUCTION

Randomised controlled trials (RCTs), that are well designed, conducted and reported, provide rigorous scientific evidence for evaluating the effectiveness and safety of

### STRENGTHS AND LIMITATIONS OF THIS STUDY

⇒ Our scoping review will use rigorous methods to identify literature using multiple sources with no restriction to regions or time periods.
⇒ The targeted review will identify recent randomised controlled trials that have used surrogate primary endpoints from six high impact journals.
⇒ Due to lack of resources for translation, we will only include records in English or Italian.
⇒ Using a purposively selected set of journals for the targeted review means that our review of recent randomised controlled trial protocols and trials is not exhaustive and may lack generalisability.

interventions intended to impact health.[1 2] Nevertheless, to meet the scientific, ethical and regulatory requirements, the conduct and delivery of RCTs is becoming increasingly resource and time-intensive,[3] with median cost estimates of up to US\$ 21.4 million for phase three trials.[4] The use of a surrogate endpoint as a substitute for a primary final patient relevant outcome[5] provides a potentially attractive solution for improving efficiency of RCTs, that is, shorter follow-up, smaller sample size, and, as a result, lower cost.

A key rationale for the use of a surrogate endpoint is that the intervention effect on the surrogate fully captures the intervention effect on the final patient relevant outcome.[6] Consideration of surrogate endpoints in RCTs has traditionally focused on the regulatory setting for pharmaceuticals and whether biomarkers are 'likely to predict' patient-centred outcomes of interest (eg, systolic blood pressure for stroke, low-density lipoprotein cholesterol for myocardial infarction, and HIV viral load for development of

AIDS). However, it is important to acknowledge a more wider application in RCTs of intermediate outcomes that are believed to capture the causal pathway through which pharmaceutical, surgical, organisational or public health interventions impact the ultimate patient-relevant outcome (eg, hospice enrolment for mortality with an intervention aimed at improving end of life care;[7] fruit and vegetable consumption for cardiovascular events for a behavioural intervention designed to improve cardiovascular risk).[8] To be regarded as a valid surrogate endpoint, a biomarker or intermediate outcome is required: (1) to reliably predict the patient/participant relevant final outcome (PRFO) in individual trial participants ('individual level' or 'patient-level' surrogacy); and (2) the intervention effect on the surrogate endpoint should reliably predict the intervention effect on the PRFO ('trial-level' surrogacy) based on evidence from meta-analyses of RCT data on both outcomes.[9 10] Statistical surrogate validation uses various statistical methods, including meta-analyses of RCT aggregate and/or individual patient data,[11 12] principal stratification,[13] causal inference,[14 15] bivariate network meta-analysis methods[16 17] and information theory.[18] However, surrogate validation should extend beyond statistical validity to include a multifaceted approach comprising of biological plausibility rationale and 'face validity' of the surrogates in trials.[19]

Despite the potential appeal of surrogate endpoints in RCTs, their use in clinical and policy decision-making remains controversial. An empirical analysis has found that RCTs using a surrogate endpoint primary outcome typically report 46% larger treatment effects compared with RCTs with final patient relevant primary outcomes.[20] This finding is supported by theoretical analyses.[21] Concerningly, some approvals based on surrogate endpoints have led to the 'real world' use of interventions that fail to demonstrate their predicted benefit(s) on the ultimate patient-centred outcome of interest and even more worryingly, that result in more harm than good.[22 23] Therefore, design and reporting of RCTs using surrogate endpoints should clearly convey the uncertainty and risks associated with their use. However, audits of RCTs to date have found this not to be the case. An analysis of 626 RCTs published in 2005 and 2006 found that 107 (17%) used a surrogate primary endpoint (surrogate endpoint as a the primary outcome) and of these, only a third discussed whether the surrogate was a valid predictor of patient-relevant outcomes.[24] Furthermore, a review of 220 cardiovascular surrogate trials found that only 59 (27%) had evidence validating the benefits of interventions on a final patient-relevant outcome.[25]

Reporting guidelines can guide design and improve the reporting of RCTs at both the protocol and report stages. Two established guidelines are as follows: SPIRIT (Standard Protocol Items: Recommendations for Interventional Trials) 2013 statement: a 33-item checklist used to guide the drafting of RCT protocols[26] and CONSORT (Consolidated Standards of Reporting Trials) 2010 Statement is a 25-item checklist used to improve reporting of

conducted trials.[27] Yet, although SPIRIT and CONSORT (and related extensions) provide general guidance on outcome reporting, there remains no standard evidence-based reference for dealing with surrogacy of the primary endpoint. Improving transparency in the reporting of trials using surrogates would enable the evidence base for the surrogate to be more effectively scrutinised. Therefore, we aim to develop extensions to report trial protocols and reports that use surrogate primary endpoints: SPIRIT-SURROGATE and CONSORT-SURROGATE, respectively. The extensions focus on trials using surrogate endpoints as primary outcomes (including as part of a composite outcome) as these would inform trial conclusions and interpretations of results and possible approval of interventions. Our working definition of a surrogate endpoint is: 'a biomarker or intermediate outcome used to substitute and predict for a final patient relevant outcome (ie, characteristic or variable that captures how a patient feels, functions or how long they survive, such as the outcomes of mortality or health-related quality of life)'.[5 6 28] Additionally, reference of surrogate endpoints in this project refer to statistically validated surrogate endpoints (eg, change in systolic blood pressure for cardiovascular mortality in antihypertensive treatments,[29 30] disease-free survival (and progression free survival in advanced disease) in colorectal cancer[31] and non-validated surrogates for which are 'reasonably likely to predict health benefit' (eg, reduction in amyloid load in Alzheimer's disease).[29 32] To develop these extensions, we will closely follow the EQUATOR (Enhancing the QUAlity and Transparency Of health Research) network's recommended steps for developing a health research reporting guideline.[33] We have structured our project into four phases: phase 1 (literature reviews), phase 2 (Delphi study), phase 3 (consensus meeting) and phase 4 (knowledge translation). This protocol outlines the activities and procedures of phase 1 consisting of scoping and targeted reviews. The scoping review will be used to: synthesise current evidence and guidance on using surrogate endpoints to generate candidate items for potential inclusion in extensions; and identify surrogate content experts for recruitment in the Delphi study (phase 2). The primary aim of the targeted review is to identify trial investigators who have led an RCT using a surrogate endpoint to be invited to participate in the Delphi exercise. A secondary aim will be to archive identified protocols and trials and use them as a 'baseline' for future evaluation of the impact of developed extensions on the reporting practice of future RCT protocols and reports.

## METHODS AND ANALYSIS
### Scoping review
The scoping review was considered to be the most suitable knowledge synthesis approach for addressing the broad aim of this study.[34] The scoping review will be conducted using a methodological framework proposed by Arksey and O'Malley,[35] and enhancements

proposed to this framework by Levac *et al*[36] and Peters *et al.*[37] This will involve six stages: formulating a research question; identifying relevant studies; inclusion of studies; charting data; summarising and reporting results and consultation.[35]

### Framework stage one: formulating the research question

This scoping review seeks to identify a list of items that should be considered when reporting RCT protocols and reports which use surrogate endpoints. Therefore, our overarching research question combines a broad scope and a specific area of inquiry[36] (ie, surrogate endpoint use): what is the current understanding, advice and guidance on using surrogate endpoints in RCTs? Specific research questions are as follows:

1. How are surrogate endpoints defined?
2. What are the limitations of using surrogate endpoints in RCTs?
3. When is the use of surrogate endpoints acceptable?
4. What published advice and guidance exists on reporting RCTs protocols and reports using surrogate endpoints?

There is a possibility of modification of these research questions during the literature reviewing and this will be reported when publishing the findings.

### Framework stage two: identifying relevant literature

We will adopt a search approach that balances comprehensiveness, breadth and feasibility.[36] Relevant literature will be identified through: (1) electronic bibliographic databases (Excerpta Medica Database (EMBASE), Medical Literature Analysis and Retrieval System Online (MEDLINE), Cochrane Methodology Register); (2) grey literature (Google and relevant website search); (3) handsearching of reference lists and (4) solicitation for additional literature from expert colleagues.[35–37]

Electronic databases and grey literature search will be supported by an experienced information specialist (VW). We have developed an initial search strategy for MEDLINE and EMBASE which combines 'surrogate endpoints', 'guidelines' and 'trials' related search terms (see online supplemental tables 1 and 2). This strategy was checked for validity against four highly cited articles (>50 citations) that answer either of our specific research questions.[20 23 24 38]

For grey literature, search strategies will be modified for each of the websites and for each strategy, the search terms and the number of results retrieved and/or screened will be recorded.[39] Online supplemental table 3 shows search strategies to be used in the Google search engine and in some of the relevant websites. Generally, the strategies will include combination of search terms (eg, 'surrogate endpoints' AND 'guidance') in Google advanced search; broad searches (eg, surrogate endpoints) using the website search function and browsing for websites without a search function. For large websites (eg, www.ema.europe.eu), Google advanced search will be used, and search limited to the website URL. The first 100 hits in each search will be screened for eligibility to balance between feasibility and relevancy of records.[39] One reviewer will screen searches on the Google search engine or websites using title and, if present, any short text underneath.

All reference lists of included full texts will be screened to identify relevant records. We will solicit for additional resources from surrogate and outcome measurement experts including authors of a recent scoping review (on 'outcome reporting recommendations for trial protocols and reports') which identified eight documents that focused on reporting recommendations for surrogate outcomes.[40]

### Framework stage three: literature selection

Databases search results will be exported to Endnote version X9 for the removal of duplicates. The remaining records will be exported to Covidence[41] for eligibility screening based on title, abstract and full-text reading by two reviewers. Title and abstract screening of grey literature will be done in respective websites by one reviewer and full-text screening done from HTML files by two reviewers.

Once full-text screening has been concluded, reviewers will hand search reference lists of all included full-texts for relevant records. The identified records combined with those supplied from experts will undergo full-text screening. Records will be eligible for inclusion if they report findings relevant to any research question. While we will mainly include records that are peer-reviewed literature, academic or regulatory grey literature (eg, white papers), reviewers will make judgements on inclusion of other records (eg, conference abstracts) based on relevance to review questions and trustworthiness of evidence presented. We will not restrict our inclusion of literature to regions or time periods. However, we will only include records in English or Italian due to lack of resources for translation. Disagreements between reviewers will be resolved by consensus or, if necessary, involving a third reviewer.

### Framework stage four: charting the findings

The following data will be extracted: author (and contact of corresponding author), publication year, country, author affiliation category (eg, academic, regulatory body, patient/public forums), record type (eg, review article, commentary and regulatory guidance), research area if specified, funding if stated and findings relevant to research questions (ie, definition, limitations, acceptability and guidance on surrogate endpoints use). A pilot will be undertaken to check if the data extraction template needs modification. All data extraction will be done by one reviewer. At the start of extraction, a subset of extracted data (~10% of records) will be checked for accuracy by a second reviewer and if accurate the first reviewer will proceed to extract in all other records.

### Framework stage five: synthesis and reporting the findings

All analysis will be done in Microsoft Excel. Descriptive data (ie, publication year, country/region, author affiliation category and record type) will be analysed using counts and percentages and presented in tables, graphs or as text. Data related to research questions (eg, key messages/advice/guidelines on surrogate endpoints use) will be collated verbatim under each research question. A simple form of thematic analysis[42] will then be used to synthesise data. Two reviewers will independently read the collated data under each research question and for each record, summarise it into: (1) item(s) to be considered when reporting protocols and trials using surrogate endpoints; and (2) whether the items are new or modifications to the SPIRIT and/or CONSORT checklist items and for new items, the section of the checklist where they should be reported. The reviewers will then meet for a virtual workshop to discuss and agree on items and their designated sections of the checklist. We will report the findings in an open-access peer reviewed publication using the Preferred Reporting Items for Systematic reviews and Meta-Analysis for Scoping Reviews.[43]

### Framework stage six: consultation exercise

The aim of consultation is to share scoping review findings with stakeholders so as to identify additional relevant resources and valuable insights that the scoping review findings may have missed.[35] Nevertheless, it is important to specify when, how and why to do consultation, the types of stakeholder involved, and how to integrate the information with review findings.[36] We will use preliminary review findings to seek insights, through virtual meetings, from Public and Patient Involvement (PPI) representatives on the identified items for reporting surrogate endpoints. Our project PPI Lead (DS) will coordinate consultation with PPI representatives, and this will offer an opportunity for knowledge transfer and exchange. Additionally, we will invite our multidisciplinary expert advisory Executive Committee members (see Acknowledgement) and the MRC-NIHR Trials Methodology Research Partnership Outcomes Working Group (www.methodologyhubs.mrc.ac.uk/), specifically the Surrogate Outcomes subgroup, to comment on any additional resources, items and perspectives not included in the preliminary findings. Review comments on the preliminary findings document or detailed notes taken during consultation meetings will be used in summarising and integrating suggested items into the review findings.

### Targeted review

The targeted reviews are intended to identify trial investigators who have led an RCT assessing a surrogate endpoint and protocols and trials that have a primary surrogate endpoint.[20] MEDLINE through PubMed will be searched for RCTs published in the last 5 years (2017–2021) in six high impact general medical journals: *Annals of Internal Medicine, BMJ, Journal of the American Medical Association, New England Journal of Medicine, Lancet* and *PLoS Medicine.*

Use of general medical journals allows for inclusion of records across a range of clinical areas. Given the focus of the project on reporting guidelines for trial protocols, we also will search two journals widely used for publishing RCT protocols: *BMJ Open* and *Trials.* We will include trial protocols and reports that use outcomes that meet our working definition of surrogate endpoints.

All identified protocols and trials will be exported to Endnote version X9 for the removal of duplicates and exported to Covidence[41] for eligibility screening. Given the primary objective of this review is to identify trial investigators who have used surrogate endpoints, screening will be limited to titles and abstracts. Two reviewers will screen all records and include those protocols and randomised trials that use surrogate primary endpoints and report intervention studies. A more in-depth screening and analysis of the full texts will be done as part of an upcoming project, acting as a baseline to evaluate the impact of the extensions (postpublication) on the reporting of RCT protocols and trials.

From the included records, one reviewer will extract the title, journal, year of publication, research area, corresponding author name, institutional affiliation, and email address. These data will be used to sample and recruit participants for the Delphi study (phase 2 of the project).

### Patient and public involvement

One of the project team members (DS) is a leading PPI advocate who has been involved in health research at local, national and international level. As outlined, PPI will be integrated in stage six of the scoping review. We are additionally exploring how patients and the public can be meaningfully involved in this project.

### Limitations

Although we will use four strategies in our scoping review searches, our inclusion will be limited to records in English and Italian language hence exclusion of non-English/Italian literature. Nevertheless, our review does not aim to be exhaustive but to identify important items for consideration when using surrogate endpoints and it is highly likely items synthesised from records in the English and Italian language would be transferable to other settings. Using an approach of a purposively selected set of journals means our targeted review of recent RCT protocols and trials is not exhaustive and may lack generalisability.

### ETHICS AND DISSEMINATION

The reviews do not require ethics approval. The reviews findings will be disseminated through conference presentations and open-access publications.

**Author affiliations**
[1]MRC/CSO Social and Public Health Sciences Unit, Institute of Health and Wellbeing, University of Glasgow, Glasgow, UK
[2]Population Health Sciences, Bristol Medical School, University of Bristol, Bristol, UK
[3]Patient and Public Involvement Lead, Nottingham, UK

⁴Edinburgh Clinical Trials Unit, Usher Institute, University of Edinburgh, Edinburgh, UK
⁵Robertson Centre for Biostatistics, Institute of Health and Well Being, University of Glasgow, Glasgow, UK
⁶SDA Bocconi School of Management, Milan, Italy

**Acknowledgements** We wish to acknowledge our dear colleague Professor Amber Young who lost her brave battle with cancer and passed away on 17th September 2022. We acknowledge our advisory Executive Committee members who will oversight the project and contribute to the consultation stage of the scoping review: Professor Joseph Ross (Chair of the Executive Committee); Professor Martin Offringa; Dr Nancy Butcher; Professor An-Wen Chan; Professor Gary Collins; Professor Sylwia Bujkiewicz; Dr Dalia Dawoud and Dr Mario Ouwens.

**Contributors** PD, CW, AY, RT and OC were involved in funding acquisition. AMM, PD, DS, CW, AY, RT and OC were involved in the initial phases of study conception and design. AMM, VW, RT and OC were involved in design of the search strategy and responsible for the first draft of the manuscript. PD, DS, CW and AY reviewed the first draft, and all authors approved the final version.

**Funding** The development of SPIRIT and CONSORT extensions has been funded by the Medical Research Council (PD, CJW, AY, RT, OC, grant number MR/V038400/1).

**Competing interests** None declared.

**Patient and public involvement** Patients and/or the public were involved in the design, or conduct, or reporting or dissemination plans of this research. Refer to the Methods section for further details.

**Patient consent for publication** Not required.

**Provenance and peer review** Not commissioned; externally peer reviewed.

**ORCID iDs**
Anthony Muchai Manyara http://orcid.org/0000-0001-6276-926X
Christopher Weir http://orcid.org/0000-0002-6494-4903
Amber Young http://orcid.org/0000-0001-7205-492X

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
