## [Reviewer comments · BMJ Open]

ARTICLE DETAILS

TITLE (PROVISIONAL)	Protocol for scoping and targeted reviews to support development of SPIRIT and CONSORT extensions for randomised controlled trials with surrogate primary endpoints
AUTHORS	Manyara, Anthony; Davies, Philippa; Stewart, Derek; Wells, Valerie; Weir, Christopher; Young, Amber; Taylor, Rod; Ciani, Oriana

VERSION 1 – REVIEW

REVIEWER	Emura, Takeshi Kurume University
REVIEW RETURNED	12-Apr-2022

GENERAL COMMENTS	This paper describes a protocol to develop a new guideline for RCTs using surrogate endpoints. The paper gives a search method (e.g. the Google search) and the documentation (e.g. by the Endnote, Microsoft Excel) to complete the documents of the guideline. The idea of the protocol is reasonable since many RCTs using surrogate endpoints appear to be ad-hoc. Some guidelines are certainly helpful to improve the relevance of the RCTs aiming at improving patients' benefit. Authors may consider three items when they revise the paper. 1. It is desirable for a surrogate endpoint to be validated statistically before it replaces the true endpoint. This validation process typically use the "correlation" between the true and surrogate endpoints. I refer the book of "The evaluation of surrogate endpoints", and [1] for this point. This important point is missing in the paper, and hence, I am afraid the use of uncorrelated surrogates in RCTs. While it may not be reasonable to impose statistically validated surrogate endpoints in all RCTs, some comments/considerations will be helpful in the process of developing the guideline. The statistical validation guarantees that a surrogate endpoint is correlated to the true endpoint at both trial-level and patient-level. [1] Green, E. M., Yothers, G., & Sargent, D. J. (2008). Surrogate endpoint validation: statistical elegance versus clinical relevance. Statistical Methods in Medical Research, 17(5), 477-486.
---

	2. Introduction could be improved in different ways.  - The word "surrogat primary endpoint" is confusing to me, and may be changed to "surrogate endpoint". When one uses the term "surrogate endpoint", there exists the "primary endpoint" or "true endpoint". Therefore, any surrogate endpoint cannot be the primary endpoint (at least in theory). Alternatively, the authors could use "surrogate endpoint as a primary endpoint". In addition, the definitions of "surrogate endpoints" and "the true endpoint" should be clearly defined without confusion. At least, I do not think "primary final patient relevant outcome" is a good word (P.4). It could be "true endpoint". 3. There are a large number of surrogate endpoints, some of them are valid and others are invalid. An example of valid surrogate endpoints is helpful for readers who are not familiar with the topic. For instance, based on my knowledge, DFS (and PFS in advanced disease) are valid surrogates for OS in colorectal cancer [2]. [2] Buyse, M., Burzykowski, T., Michiels, S., & Carroll, K. (2008). Individual-and trial-level surrogacy in colorectal cancer. Statistical Methods in Medical Research, 17(5), 467-475.
--	---

REVIEWER	Campbell, Michael University of Sheffield, Health Services Research SchARR
REVIEW RETURNED	25-Jun-2022

GENERAL COMMENTS	1) I don't have much to say about this protocol, except to ask whether they will include pilot studies in their review. In my experience some trials are described as a pilot trial because they use a surrogate endpoint, but I have argued this is misleading¹ and they are in fact not pilot studies unless they clearly are committed to a subsequent main trial. 2) It is good that Italian journals are also considered. It would be interesting to take advantage of this to see if there are any differences between English language and Italian reports. I assume CONSORT is available in Italian? Ref 1. Campbell, M.J., Lancaster, G.A. and Eldridge, S.M., (2018). A randomised controlled trial is not a pilot trial simply because it uses a surrogate endpoint. Pilot and Feasibility Studies,4(1), p.130 Minor comments In references for papers it is usual to have journal name in italics and not the article titles Ref 6 BMJ: British Medical Journal? Page 8 line 48 'in an open-access'
---

VERSION 1 – AUTHOR RESPONSE

Reviewer: 1 Dr. Takeshi Emura, Kurume University

The idea of the protocol is reasonable since many RCTs using surrogate endpoints appear to be ad-hoc. Some guidelines are certainly helpful to improve the relevance of the RCTs aiming at improving patients' benefit. Authors may consider three items when they revise the paper. 1. It is desirable for a surrogate endpoint to be validated statistically before it replaces the true endpoint. This validation process typically use the "correlation" between the true and surrogate endpoints. I refer the book of "The evaluation of surrogate endpoints", and [1] for this point. This important point is missing in the paper, and hence, I am afraid the use of uncorrelated surrogates in RCTs. While it may not be reasonable to impose statistically validated surrogate endpoints in all RCTs, some comments/considerations will be helpful in the process of developing the guideline. The statistical validation guarantees that a surrogate endpoint is correlated to the true endpoint at both trial-level and patient-level. [1] Green, E. M., Yothers, G., & Sargent, D. J. (2008). Surrogate endpoint validation: statistical elegance versus clinical relevance. Statistical Methods in Medical Research, 17(5), 477-486.	Thank you. We agree that use of statistically validated surrogates increases certainty of health benefit observed in trials. We are confident we will pick up this point in our scoping review data extraction/synthesis and present it to Delphi participants for rating and if there is consensus include it in the guidelines. We also agree that it may not be reasonable to impose validated surrogates in all trials. For example the FDA current listing includes surrogates that have previously used in their approvals but not necessarily been statistically validated. Hence, we have clarified that we are interested in both validated and non-validated surrogates for which there is still not convincing evidence that they are reasonably likely to predict benefit.	We have added the following on page 5, second paragraph: Additionally, reference of surrogate endpoints refers to statistically validated surrogate endpoints (e.g., change in systolic blood pressure for cardiovascular mortality in anti-hypertensive treatments) and non-validated surrogates for which there is still not convincing evidence that they are 'reasonably likely to predict health benefit' (e.g., reduction in amyloid load in Alzheimer's disease)
2. Introduction could be improved in different ways.  - The word "surrogate primary endpoint" is confusing to me and may be changed to "surrogate endpoint". When one uses the term "surrogate endpoint", there exists the "primary endpoint" or "true endpoint". Therefore, any surrogate endpoint cannot be the primary endpoint (at least in theory). Alternatively, the authors could use "surrogate endpoint as a primary endpoint". In	Thank you for this observation. Our completed SPIRIT CONSORT-SURROGATE guidelines will target trials whose primary outcome is a surrogate endpoint and so we needed to be clear of that context. We have clarified this on the first time use of "surrogate primary endpoint".	We have defined a surrogate primary endpoint on first use of the term: surrogate primary endpoints (surrogate endpoint as the primary outcome) on the abstract on page 2 and introduction on page 4. We have also added the following statement to rationalise focus of surrogate endpoints when

addition, the definitions of "surrogate endpoints" and "the true endpoint" should be clearly defined without confusion. At least, I do not think "primary final patient relevant outcome" is a good word (P.4). It could be "true endpoint".	We have defined surrogate endpoints as biomarkers and intermediate outcomes that substitute for and predict for a final patient/participant relevant outcome (i.e., characteristic or variable that captures how a patient feels, functions, or how long they survive, such as the outcomes of mortality or health-related quality of life) on page 5, second paragraph We acknowledge that there isn't consensus on reference and definitions of these terms including "true endpoints" hence one of our research questions is to explore how surrogate endpoints are defined.	used as primary outcomes on page 5: The extensions focus on trials using surrogate endpoints as primary outcomes (including as part of a composite outcome) as these would inform trial conclusions and interpretations of results and possible approval of interventions
3. There are a large number of surrogate endpoints, some of them are valid and others are invalid. An example of valid surrogate endpoints is helpful for readers who are not familiar with the topic. For instance, based on my knowledge, DFS (and PFS in advanced disease) are valid surrogates for OS in colorectal cancer [2]. [2] Buyse, M., Burzykowski, T., Michiels, S., & Carroll, K. (2008). Individual-and trial-level surrogacy in colorectal cancer. Statistical Methods in Medical Research, 17(5), 467-475.	Thank you for this suggestion. We have now clarified after the definition that reference to surrogate endpoints refers to both validated and non-validated surrogates reasonable likely to predict benefit and given examples of both citing Fleming and Powers Fleming TR, Powers JH. Biomarkers and surrogate endpoints in clinical trials. Stat Med. 2012 Nov 10;31(25):2973-84	We have added the following on page 5, second paragraph: Additionally, reference of surrogate endpoints refers to statistically validated surrogate endpoints (e.g., change in systolic blood pressure for cardiovascular mortality in anti-hypertensive treatments) and non-validated surrogates for which there is still not convincing evidence that they are 'reasonably likely to predict health benefit' (e.g., reduction in amyloid load in Alzheimer's disease)
Reviewer: 2 Dr. Michael Campbell, University of Sheffield		
1) I don't have much to say about this protocol, except to ask whether they will include pilot studies in their	Thank you for this observation. Yes, we will include all trials in the	No change

review. In my experience some trials are described as a pilot trial because they use a surrogate endpoint, but I have argued this is misleading and they are in fact not pilot studies unless they clearly are committed to a subsequent main trial. 1. Campbell, M.J., Lancaster, G.A. and Eldridge, S.M., (2018). A randomised controlled trial is not a pilot trial simply because it uses a surrogate endpoint. Pilot and Feasibility Studies,4(1), p.130	targeted review. We may or may not find such pilot studies in the six high impact general medical journals that we aim to use.	
2) It is good that Italian journals are also considered. It would be interesting to take advantage of this to see if there are any differences between English language and Italian reports. I assume CONSORT is available in Italian?	Not Italian journals per se but Italian language records that come up in our search would be included unlike other languages. Should we find any Italian reports we will compare to see if there are differences. Yes, CONSORT is available in Italian	No change
Minor comments In references for papers it is usual to have journal name in italics and not the article titles Ref 6 BMJ: British Medical Journal? Page 8 line 48 'in an open-access'	Thank you for these suggested edits. We have changed to BMJ reference style that allows for journal name to be in italics	Page 8, line 48 now reads 'in an open-access'. Reference style changed.

VERSION 2 – REVIEW

REVIEWER	Emura, Takeshi Kurume University
REVIEW RETURNED	Kurume University 06-Aug-2022

GENERAL COMMENTS	The paper was revised by authors. However, I could not see any serious consideration on the issues that I raised. Opinions for the revised paper for my main concerns 1) About statistically validated surrogate endpoints: The change made on this concern is superfluous; the addition of a few sentences without any reference or serious discussions for statistical validation methods. I provided two important references for authors to explore the importance of statistical validations on
--

	surrogates and its clinical relevance. I regret to see they are simply ignored. 2) I simply replicate my concern again since I did not see any change. 3) There are not serious considerations for the issues I raised and about the reference I mentioned.
--	--

REVIEWER	Campbell, Michael University of Sheffield, Health Services Research SchARR
REVIEW RETURNED	06-Sep-2022

GENERAL COMMENTS	Thank you for considering my suggestions
--

VERSION 2 – AUTHOR RESPONSE

Reviewer comments	Authors response	Modification to manuscript
Comments from the Reviewer:		
The paper was revised by authors. However, I could not see any serious consideration on the issues that I raised. Opinions for the revised paper for my main concerns 1) About statistically validated surrogate endpoints: The change made on this concern is superfluous; the addition of a few sentences without any reference or serious discussions for statistical validation methods. I provided two important references for authors to explore the importance of statistical validations on surrogates and its clinical relevance. I regret to see they are simply ignored. 2) I simply replicate my concern again since I did not see any change. 3) There are not serious considerations for the issues I raised and about the reference I mentioned.	Thank you for reviewing our revised manuscript. We agree that whilst our project does not address statistical validation it is important to have a summary of statistical methods used. We have therefore added literature to this effect (11 references) including the two references that you suggested. We provide a point-by-point response to comments from the first peer review, see below	The following statements have now been added in page 4, last paragraph: To be regarded as a valid surrogate endpoint, a biomarker or intermediate outcome is required: 1) to reliably predict the PRFO in individual trial participants ('individual level' or 'patient-level' surrogacy); and 2) the intervention effect on the surrogate endpoint should reliably predict the intervention effect on the PRFO ('trial-level' surrogacy) based on evidence from meta-analyses of RCT data on both outcomes^{9 10}. Statistical surrogate validation uses various statistical methods, including meta-analyses of RCT aggregate and/or individual patient data^{11 12}, principal stratification¹³,

		causal inference^{14 15}, bivariate network meta-analysis methods^{16 17} and information theory¹⁸. However, surrogate validation should extend beyond statistical validity to include a multifaceted approach comprising of biological plausibility rationale and “face validity” of the surrogates in trials¹⁹.
Comments from first peer review		
The idea of the protocol is reasonable since many RCTs using surrogate endpoints appear to be ad-hoc. Some guidelines are certainly helpful to improve the relevance of the RCTs aiming at improving patients' benefit. Authors may consider three items when they revise the paper. 1. It is desirable for a surrogate endpoint to be validated statistically before it replaces the true endpoint. This validation process typically use the "correlation" between the true and surrogate endpoints. I refer the book of "The evaluation of surrogate endpoints", and [1] for this point. This important point is missing in the paper, and hence, I am afraid the use of uncorrelated surrogates in RCTs. While it may not be reasonable to impose statistically validated surrogate endpoints in all RCTs, some comments/considerations will be helpful in the process of developing the guideline. The statistical validation guarantees that a surrogate endpoint is correlated to the true endpoint at both trial-level and patient-level. [1] Green, E. M., Yothers, G., & Sargent, D. J. (2008). Surrogate	Thank you, please see our response above.	We have now added 11 references on statistical validation methods including the reference you have provided, on last paragraph on page 5: To be regarded as a valid surrogate endpoint, a biomarker or intermediate outcome is required: 1) to reliably predict the PRFO in individual trial participants ('individual level' or 'patient-level' surrogacy); and 2) the intervention effect on the surrogate endpoint should reliably predict the intervention effect on the PRFO ('trial-level' surrogacy) based on evidence from meta-analyses of RCT data on both outcomes^{9 10}. Statistical surrogate validation uses various statistical methods, including meta-analyses of RCT aggregate and/or individual patient data^{11 12}, principal stratification¹³,

endpoint validation: statistical elegance versus clinical relevance. Statistical Methods in Medical Research, 17(5), 477-486.		causal inference^{14 15}, bivariate network meta-analysis methods^{16 17} and information theory¹⁸. However, surrogate validation should extend beyond statistical validity to include a multifaceted approach comprising of biological plausibility rationale and “face validity” of the surrogates in trials¹⁹.
2. Introduction could be improved in different ways.  - The word "surrogate primary endpoint" is confusing to me and may be changed to "surrogate endpoint". When one uses the term "surrogate endpoint", there exists the "primary endpoint" or "true endpoint". Therefore, any surrogate endpoint cannot be the primary endpoint (at least in theory). Alternatively, the authors could use "surrogate endpoint as a primary endpoint". In addition, the definitions of "surrogate endpoints" and "the true endpoint" should be clearly defined without confusion. At least, I do not think "primary final patient relevant outcome" is a good word (P.4). It could be "true endpoint".	Thank you for this observation. Our response to this remains the same as earlier responded to: that the completed SPIRIT CONSORT-SURROGATE guidelines will target trials whose primary outcome is a surrogate endpoint and so we needed to be clear of that context. We have clarified this on the first time use of “surrogate primary endpoint”. We have defined surrogate endpoints as biomarkers and intermediate outcomes that substitute for and predict for a final patient/participant relevant outcome (i.e., characteristic or variable that captures how a patient feels, functions, or how long they survive, such as the outcomes of mortality or health-related quality of life) on page 5, second paragraph We acknowledge that there isn't consensus on reference and definitions of these terms including “true endpoints” hence one of our research	We have defined a surrogate primary endpoint on first use of the term: surrogate primary endpoints (surrogate endpoint as the primary outcome) on the abstract on page 2 and introduction on page 4. We have also added the following statement to rationalise focus of surrogate endpoints when used as primary outcomes on page 5: The extensions focus on trials using surrogate endpoints as primary outcomes (including as part of a composite outcome) as these would inform trial conclusions and interpretations of results and possible approval of interventions

	questions is to explore how surrogate endpoints are defined.	
3. There are a large number of surrogate endpoints, some of them are valid and others are invalid. An example of valid surrogate endpoints is helpful for readers who are not familiar with the topic. For instance, based on my knowledge, DFS (and PFS in advanced disease) are valid surrogates for OS in colorectal cancer [2]. [2] Buyse, M., Burzykowski, T., Michiels, S., & Carroll, K. (2008). Individual-and trial-level surrogacy in colorectal cancer. Statistical Methods in Medical Research, 17(5), 467-475.	Thank you for this suggestion. We have now clarified after the definition that reference to surrogate endpoints refers to both validated and non-validated surrogates that are reasonably likely to predict benefit and given examples of both citing: Fleming TR, Powers JH. Biomarkers and surrogate endpoints in clinical trials. Stat Med. 2012 Nov 10;31(25):2973-84; Buyse M, Burzykowski T, Michiels S, et al. Individual- and trial-level surrogacy in colorectal cancer. Stat Methods Med Res 2008;17(5):467-75; FDA. Table of Surrogate Endpoints That Were the Basis of Drug Approval or Licensure 2022; FDA-NIH Biomarker Working Group. BEST (Biomarkers, endpoints, and other tools) resource	We have added the following on page 5-6, second paragraph: Additionally, reference of surrogate endpoints in this project refers to statistically validated surrogate endpoints (e.g., change in systolic blood pressure for cardiovascular mortality in anti-hypertensive treatments^{29 30}, disease-free survival (and progression free survival in advanced disease) in colorectal cancer³¹) and non-validated surrogates for which are 'reasonably likely to predict health benefit' (e.g., reduction in amyloid load in Alzheimer's disease)^{29 32}

VERSION 3 – REVIEW

REVIEWER	Emura, Takeshi Kurume University
REVIEW RETURNED	29-Sep-2022

GENERAL COMMENTS	The paper is improved substantially and certainly better than the original one. I believe the paper is publishable subject to minor considerations. I am still keeping my original concern ["At least, I do not think "primary final patient relevant outcome" is a good word. It could be "true endpoint"]. I wonder if this long word in P.4 is common in the literature, and could be replaced by a shorter one, such as the true endpoint or the final endpoint. I could not see any clear response for my concern in the responses. Finally, the abbreviation "PRFO" in P.4 could be clarified (I could not see any explanation about it). However, the final decision on the above considerations is up to the authors.
--